Random k conditional nearest neighbor for high-dimensional data

Lu Jiaxuan
Gweon Hyukjun hgweon@uwo.ca
University of Western Ontario , London , ON , Canada
Arnaiz-González Álvar
Electronic publication date: 2025 Jan 24
Publication date: 2025
Volume: 11
Electronic Location ID: e2497
Received 2024 Jul 23; Accepted 2024 Oct 18
Copyright: ©2025 Lu and Gweon
Copyright year: 2025
Copyright holder: Lu and Gweon
License: This is an open access article distributed under the terms of the Creative Commons Attribution License, which permits unrestricted use, distribution, reproduction and adaptation in any medium and for any purpose provided that it is properly attributed. For attribution, the original author(s), title, publication source (PeerJ Computer Science) and either DOI or URL of the article must be cited.
License URL: https://creativecommons.org/licenses/by/4.0/

Keywords: K nearest neighbor, High-dimensional data, Nonparametric classification

Funding: The Natural Sciences and Engineering Research Council of Canada (NSERC) 04698 This work was supported by the Natural Sciences and Engineering Research Council of Canada (NSERC, grant numbers 04698 (PI: Hyukjun Gweon)). The funders had no role in study design, data collection and analysis, decision to publish, or preparation of the manuscript.

==============================
The k nearest neighbor (kNN) approach is a simple and effective algorithm for classification and a number of variants have been proposed based on the kNN algorithm. One of the limitations of kNN is that the method may be less effective when data contains many noisy features due to their non-informative influence in calculating distance. Additionally, information derived from nearest neighbors may be less meaningful in high-dimensional data. To address the limitation of nearest-neighbor based approaches in high-dimensional data, we propose to extend the k conditional nearest neighbor (kCNN) method which is an effective variant of kNN. The proposed approach aggregates multiple kCNN classifiers, each constructed from a randomly sampled feature subset. We also develop a score metric to weigh individual classifiers based on the level of separation of the feature subsets. We investigate the properties of the proposed method using simulation. Moreover, the experiments on gene expression datasets show that the proposed method is promising in terms of predictive classification performance.

Introduction

Supervised learning is one of the fundamental parts of machine learning that studies the relationship between input feature variables and a target output variable. It is employed across various fields to enhance decision-making efficacy through algorithms that learn informative patterns from data. Among the numerous supervised learning methods, the k nearest neighbor (kNN) (Fix & Hodges, 1989) method is one of the most widely used (Singh, Thakur & Sharma, 2016; Ray, 2019; Sarker, 2021). This method classifies a new instance using local information around the target query. Despite its simplicity, the kNN method has been successfully used in many applications including economic forecasting (Imandoust & Bolandraftar, 2013), disease diagnosis (Sarkar & Leong, 2000), and forestry (Chirici et al., 2016). In addition to the original version of kNN, there are variation approaches that modify aspects such as the distance metric, the definition of the neighborhood, or the classification rule. For example, the weight-adjusted kNN (Han, Karypis & Kumar, 2001) assigns weights to attributes according to the distance and determines the extent to which an attribute influences the classification. Discriminant adaptive nearest neighbor (Tibshirani & Hastie, 1996) proposed a local linear discriminant analysis for computing neighborhoods and could be employed in any neighborhood-based classifiers. Adaptive kNN (Sun & Huang, 2010) computes an optimal value of k for each test data using its nearest neighbor. Probabilistic nearest neighbor (Holmes & Adams, 2002) proposed a probabilistic framework that accommodates uncertainty in k as well as in the strength of the interaction between neighbors.

Most nearest-neighbor-based approaches utilize all feature variables as input for classification. However, real-world data may include a considerable number of features, many of which are often irrelevant to the outcome variable. The existence of non-informative features can be problematic for classification since the noisy features will dilute and mislead correct information, resulting in a significant increase in classification error (Clarke et al., 2008). This issue often occurs in high-dimensional data such as gene data (Johnstone & Titterington, 2009). Additionally, the curse of dimensionality may render distance-based information, which is crucial for the use of nearest-neighbor approaches, less effective in high-dimensional data (Verleysen & François, 2005; Nettleton, Orriols-Puig & Fornells, 2010; Beyer et al., 1999).

The ensemble method, which aggregates results from multiple classifiers to make the final decision, has gained much interest over the past three decades (Piao et al., 2014; Moon et al., 2007). To effectively classify high-dimensional data using kNN-based methods, many researchers have applied the ensemble technique with kNN. Voting kNN (Grabowski, 2002) utilizes multiple kNN classifiers with different values of k for voting the final result. Zhou & Yu (2005) proposed BagInRand (Bagging in Randomness), a bagging variant that constructs component kNN classifiers by altering both the training set and distance metric. Li, Harner & Adjeroh (2014) developed an ensemble of kNN classifiers with randomly sampled features. Gul et al. (2018) validates and ranks the component classifiers by bootstrapping from the training set. Park & Kim (2015) proposed a feature selection method based on nearest-neighbor ensemble classifiers. These techniques aim to refine kNN for high-dimensional data by utilizing and aggregating multiple classifiers.

In this article, we propose a novel nearest-neighbor-based method for high-dimensional data. Our method is based on the k conditional nearest neighbor algorithm (kCNN) (Gweon, Schonlau & Steiner, 2019), which efficiently provides class probability as well as classification results. We extend kCNN for high-dimensional data by applying it to a number of random subsets of features. Furthermore, our method further weighs the random subsets based on their importance. The performance and effectiveness of the proposed method are evaluated through simulation and experimental studies on a set of gene data.

The rest of this article is organized as follows. We introduce the proposed methods and techniques in the ‘Method’ section. We conduct simulation studies for an extensive analysis of the proposed method in different parameter settings in the ‘Simulation’ section. In ‘Experimental study on gene expression data’, we apply our method to a set of gene expression data and compare its predictive performance with other competitors. Finally, in the ‘Conclusion’ section, we conclude the article.

Method

In this section, we describe the proposed method and discuss the model parameters.

Random k conditional nearest neighbor

The random k conditional nearest neighbor (RkCNN) method combines kCNN classifiers that are used in different random feature subsets. Among the q features, a feature subset of size m (1 ≤ m ≤ q) is created using simple random sampling without replacement from the data feature space F. Each component kCNN model is then trained on one of these subsets.

Let Y be the output variable with L classes and y be an observed value of Y. Let X be the n × q feature matrix consisting of n input vectors each of length q. Consider a random feature subset Fj⊆F where the elements of Fj are selected features. The matrix corresponding to the random subset Fj is denoted by

Xj′=x′1T⋮x′nT,

where xi′ is the feature vector of the ith observation with m features selected for Fj.

When predicting the class of a new instance, the estimated probabilities are calculated by applying kCNN to the feature subset. In particular, for a given query vector x, kCNN looks up the k-th nearest neighbor from each class and assigns the corresponding class probabilities (1) P ˆjY=c|x=x′,xk|c′2−1 ∑l=1Lx′,xk|l′2−1

where ⋅2 is the L2-norm, and xk|c′ represents the k-th nearest neighbor of x′ from class c.

Applying Eq. (1) to all h random feature subsets results in h sets of distinct class probability estimates. The proposed method aggregates these class probability estimations using weighted averaging, where the weights depend on the level of informativeness of the feature subsets for classification. If none of the features in a random subset plays an important role in classification, class probability estimates obtained from this subset space will add noisy information, potentially deteriorating the classification performance. Consequently, the proposed method assigns weights to estimated class probabilities from individual classifiers based on the informative level of their respective subset spaces. Specifically, we quantify the informativeness level for each feature subset using the separation score (for a detailed description of the separation score, please refer to ‘Model Parameters’) and compute the weights accordingly.

Given a set of separation scores S1, …, Sh, we sort the separation score in descending order and denote them using order statistics, such as S(1), S(2)…S(h). For effective weighting, the RkCNN algorithm disregards results from noisy feature subset spaces with relatively low separation scores. The weights are then computed using the relative magnitudes of the remaining scores. Specifically, we only consider the class probability results from the r subsets that correspond to the top r separation scores, denoted as S(1), …, S(r). Based on these r remaining scores, the weights for each classifier are computed as

wj=Sj ∑l=1rSl.

RkCNN estimates the aggregated probability P ˆY=c|x= ∑j=1rP ˆjY=c|x⋅wj

where P ˆj is the estimated probability obtained in the feature subset corresponding to S(j). For the classification task, RkCNN chooses the class with the greatest probability. That is, Y ˆ= argmaxcP ˆY=c|x.

Algorithm 1 summarises the procedure of RkCNN. We also illustrate the procedure of RkCNN in Fig. 1. The tuning parameters for RkCNN are discussed in ‘The number of nearest neighbors: k’.

Figure 1 Illustration of the RkCNN algorithm.

__________________________________________________________________________________________________________ Algorithm 1 RkCNN   1:  for j = 1 to h do  2:       Sample m features from the feature space F, denoted as Fj, where Fj ⊆   F  3:       Calculate the corresponding Separation Score Sj =  BV WV   4:  end for  5:  Sort {Sj}hj=1  in descending order, denoting the sorted values as S(j), i.e.,      S(1) ≥ S(2) ≥⋅⋅⋅≥ S(h)   6:  Sort Fj by S(j), denoted as F(j)   7:  for j = 1 to r do  8:       Use the matrix of F(j), X′(j), and y to construct a kCNN model, denote      as kCNN(j)   9:       Calculate weights w(j) =     S(j) _____∑ r     l=1 S(l) 10:       Calculate ˆ P(j)(Y = c|x), the predicted probability of class c by kCNN(j) 11:  end for 12:  Integrated    probability    for    class    c   is,     ˆ P(Y     =      c|x)      =      ∑r         j=1 [        ˆ P(j)(Y = c|x) ⋅ w(j)] 13:  Classify x to class c which maximise  ˆ P(Y = c|x) _________________________________________________________________________________________________________________

Separation score

The proposed RkCNN method assigns different weights to individual feature subsets. Utilizing noisy feature subsets that contain irrelevant features does not enhance model performance, it is advisable to assign small weights to such subsets. If a subset consists of features irrelevant to the classification task, no pattern is expected between the set of selected features and the class label. Conversely, kCNN will perform well in a subset in which the classes are well-separated.

To quantify the level of discriminative information available in any given feature subset, we use the between-group and within-group variances. Specifically, the variance between groups is defined as the variance among the center points of each class.

Let x ¯c be the mean vector for the class c and x ¯ be the overall mean vector. For a feature subset Fj, the between-group variance (BV) is the variance of L center points BV=∑c=1Lx ¯c′−x ¯′2L−1.

The variance within groups is the mean of the variances in each class. Let xic be the ith observations in class c, where ic = 1, …, Nc. The variance for class c is Varc=∑ic=1Ncxic′−x ¯c′2Nc−1

where Nc is the number of data points in class c.

The variance within groups is the average of L variances WV= ∑c=1LVarcL= ∑c=1L ∑ic=1Ncxic′−x ¯c′2Nc−1L.

Then, the separation score for the subset is defined as S=BVWV=∑i=cLx ¯c−x ¯′2L−1/∑c=1L ∑ic=1Ncxic′−x ¯c′2Nc−1L.

A small S value indicates a relatively low BV compared to WV, suggesting that the class means are close to the overall mean with larger within-class variances. In such cases, the class means are relatively close to the overall mean and each class has a relatively larger variance. Hence, the class probability estimated by kCNN will be barely meaningful. Conversely, a large S value suggests that the feature subset’s BV is relatively large in comparison to its WV. This indicates that the class means are distant from the overall mean, and the data is well-separable within this feature subspace. Such a feature subset is highly advantageous for utilizing kCNN, as it allows the classifier to effectively distinguish between classes.

Note that the computation of separation scores is model-free, as the separation score does not require the implementation of the kCNN. This implies that the computation can be quick once the following two matrices have been calculated. B=b1⋯bL=x ¯1−x ¯∘x ¯1−x ¯⋯x ¯L−x ¯∘x ¯L−x ¯

W=w1⋯wN=x1−x ¯1∘x1−x ¯1N1−1⋯xN−x ¯L∘xN−x ¯LNL−1

where ∘ is the element-wise product, bc and wi represent the c-th and ith column vectors of B and W, respectively.

For a feature subset Fj, let zj = (z1j, z2j…, zqj) be the indicator vector, where zij takes the value 1 if ith feature is in Fj and 0 otherwise. Then BV and WV can be obtained by (2) BV=∑c=1LbcTzjL−1andWV=∑i=1NwiTzjL.

The result Eq. (2) implies that once matrices B and W are obtained, the computation of BV and WV (and thus the separation score) for individual feature subsets can be quick because no additional computations for individual vectors in B and W are needed.

While RkCNN generates h random subsets, the method only uses the subsets with the top r (≤h) separation scores. Assuming that the number of informative features follows a hypergeometric distribution, in the presence of many non-informative features, most subsets may contain only a few informative features. This implies that the distribution of the separation score will likely be positively skewed. Using the top r subsets helps to block the contribution of non-informative subsets.

Selecting the top r subsets is analogous to filter methods in feature selection. While filter methods select top-ranked individual features, RkCNN ranks random subsets based on their separation scores.

It is worth noting that the proposed separation score is related to Fisher’s linear discriminant analysis (LDA) (Fisher, 1936) which measures the level of separation using the variances between and within the classes. Fisher’s LDA projects the data points onto a one-dimensional space such that the projections have large between-group variance and small within-group variance. Although both Fisher’s LDA and our separation score use the between-/within-class variances, Fisher’s LDA is designed to identify the vector that maximizes the ratio of between-group variance to within-group variance. In contrast, we use the separation score to quantify the level of separability in random feature subsets.

Model parameters

In this section, we discuss the model parameters of RkCNN.

The number of nearest neighbors: k

As with other kNN-based approaches, the performance of RkCNN is dependent on the choice of k. Additionally, the parameter k serves as an adjuster for the bias–variance trade-off (Hastie, Tibshirani & Friedman, 2009). A small value of k yields a model with a large variance and low bias. The prediction highly relies on local patterns from a small number of nearest observations, making it more flexible to the training data and reducing bias. However, this flexibility also renders the model sensitive to noise or minor changes, leading to higher variance. In contrast, a large value of k results in a model with a large bias and low variance. It averages over a greater number of neighbors for predictions. Although this oversimplification may overlook complicated patterns, it enhances stability across datasets, reducing variance and weakening the impact of minor changes in the training data.

The proposed RkCNN method takes the average of many kCNN models from different feature subsets. In the context of the bias–variance trade-off, this averaging technique helps reduce the variance without affecting the bias (Gupta et al., 2022). This implies that RKCNN is likely most effective when individual kCNN models have large variance but small bias. Consequently, we expect that RkCNN will perform well at small values of k.

The number of features in any feature subset: m

The parameter m represents the number of features selected for each random subset within the RkCNN framework. Increasing m enhances the likelihood of a feature being sampled for a random subset. However, excessively large values of m may lead to substantial overlap among the features in different random subsets, resulting in similar subset structures and correlated performance across these models. Conversely, the main concern with small values of m is that the resulting model may overlook interactions between the features that are potentially useful for classification. In the extreme case where m = 1, selecting the top r subsets is equivalent to selecting the top r features based on their separation scores. Consequently, these separation scores reflect only the marginal contributions of individual features, without accounting for any interactions with other features. Thus, an appropriate value of m should balance the exploitation of feature interactions while maintaining a low level of overlap between the random subsets.

The number of contributing subsets: r

The parameter r represents the number of kCNN models that contribute to the RkCNN prediction. Increasing r is intended to enhance model robustness and model efficiency by incorporating a diverse set of models, each trained on a random subset of features. This diversity helps to reduce the risk of overfitting and improves the generalization of the model.

As RkCNN bases its predictions on the average of r model outcomes, the influence of any individual classifier diminishes as r increases, leading to a more stable prediction. This process allows the ensemble to capture more diverse patterns and finer details in the data, typically improving the overall accuracy. However, incorporating additional kCNN models increases the runtime and computational load. Therefore, while a higher r is desirable for improving model performance, it must be balanced against the computational costs.

Optimally, RkCNN performs best at higher r values, provided that the computational overhead remains within an acceptable range. This balance ensures that RkCNN remains both effective and efficient, making it suitable for various applications where robustness and quick prediction times are critical.

The total number of sampled random subsets: h

The parameter h (≥r) represents the total number of feature subsets generated by the RkCNN. Based on the separation scores, the RkCNN method only uses the top r out of h subsets for prediction. The remaining (h − r) subsets are considered noisy subsets and thus excluded, as they do not significantly contribute to classification outputs. This selective approach allows RkCNN to focus computational resources on the most informative subsets, enhancing the efficiency and effectiveness of the model.

It is important to note that the subset filtering procedure is conducted prior to the construction of kCNN classifiers. That is, increasing h does not affect the number of kCNN classifiers ultimately constructed, thereby having a minimal impact on the computational time of the RkCNN algorithm by variations in h.

In summary, we expect that RkCNN performs well at a small value of k, a relatively small to moderate value of m, and a large value of r. Lastly, h can be any integer greater than or equal to r, and we examine h = r, 2r…, 10r in the ‘Simulation and Experimental Study on Gene Expression Data’ section.

Simulation

In this section, we explore the performance of the RkCNN model through a series of simulations. The simulation experiments are designed with various tunable parameters that allow us to systematically investigate the model’s behavior in environments with different levels of noisy features.

Each simulation setting is crafted to mimic realistic scenarios where the presence of noise can significantly impact classification accuracy. Through these experiments, we aim to provide a comprehensive evaluation of the RkCNN model, focusing on its effectiveness in handling diverse and challenging data landscapes.

Data generation

Following the methodology outlined in Gul et al. (2018), the informative features of the two classes are generated with correlated and uncorrelated structures respectively. Suppose there are d (≤q) informative features in the dataset. For class A, the feature variables are generated from a multivariate normal distribution NμA,ωΨ, where the mean vector μA is uniform with each element set to 2. Here, ω is a scaling factor that controls the overall variance of the features, modulating the extent of spread within class A. The variance–covariance matrix Ψ, a d × d matrix, is defined as Ψ=σ1,1σ1,2⋯σ1,dσ2,1σ2,2⋯σ2,d⋯⋯⋯⋯σd,1σd,2⋯σd,d

where σi,j=1/2i−j. Conversely, the feature variables for class B are generated independently from NμB,Id, where the mean vector μB has value 1 on all elements and Id is the d × d identity matrix, leading to uncorrelated features.

In each class, features are generated from identical distributions, ensuring that informativeness and separability are uniformly controlled. Then, non-informative features are generated from an independent standard normal distribution to complete the dataset.

Additionally, to explore scenarios involving multiple classes, we extend the data generation scheme to include a third class, C. Instances for class C are generated from NμC,ωCId, where the mean vector μC has value 3 on all elements. ωC is the variance parameter specific to class C, controlling the dispersion of feature values in Class C and influencing the degree of overlap with other classes, thereby affecting the classification challenge.

In the remainder of this section, we conduct a comprehensive examination of the performance of the proposed method across various simulation settings. This involves varying the data size, the number of non-informative features, the number of classes, and the value of ω.

Simulation results and analysis

Separation score vs number of informative features

We evaluate the proposed separation score as an informativeness measure of feature subsets. The relationship between the separation score and the number of informative features is illustrated in Fig. 2. For each subplot within the figure, the simulation data contained 1,000 instances per class, comprising 20 informative features and 500 non-informative features as noisy factors. We analyzed the distribution of 800 separation scores.

Figure 2 The distribution of separation scores under different simulation settings.

The value of m represents the number of features in random subsets.

The result indicated that the separation score for a feature subset increased as the subset contained more informative features. Notably, the most significant increase in score occurred when moving from zero to one informative feature, i.e., transitioning from a subset of purely noisy features to one containing an informative element. This pattern held true in both binary and multi-class scenarios, emphasizing the separation score as a potential proxy for the unknown count of informative features in practical applications.

Effect of k and m

Figure 3 illustrates the number and proportion of informative features within the r = 200 contributing subsets across varying values of m. The left panel showed a positive correlation between m and the number of informative features. As more features were included in a random subset, the likelihood of sampling informative features increased. Conversely, the right panel revealed a negative correlation between m and the proportion of the informative features. This indicated that larger subsets tended to be more dominated by noisy features, diluting the concentration of informative elements.

Figure 3 Number (left) and proportion (right) of informative features among the contributing subsets.

Figure 4 shows RkCNN’s classification accuracy across different values of m and k in the binary and multi-class problems. We varied ω and ωC to adjust the extent of spread in the classes A and C, affecting their degree of overlap. When the classes were well-separated (i.e., ω = 1 for the binary and ω = ωC = 1 for the multi-class problems), neither k nor m significantly affected RkCNN’s accuracy. However, as the class overlap increased with higher ω and ωC, the performance of RkCNN became sensitive to the choice of k and m. The decreased accuracy at higher m values was attributed to the raised proportion of non-informative features, as depicted in the right panel of Fig. 3, which compromised the precision of class probability estimations. Notably, smaller values of k generally performed well regardless of m in the majority of cases. Our result regarding the relationship between k and the level of class overlap is also consistent with that of García, Mollineda & Sánchez (2008) who use the kNN classifier. Based on these findings, smaller values of k and m were recommended to optimize the classification performance of RkCNN, particularly under conditions of significant class overlap.

Figure 4 RkCNN classification accuracy for different k and m.

Effect of r and h at different noisy levels

Figure 5 shows the accuracy of RkCNN across different values of r and h under various noisy conditions, with the number of non-informative features ranging from 100 to 1,000. We fix k = 1 and m=⌈q⌉ for this analysis.

Figure 5 RkCNN with r and h at different noisy levels.

The results demonstrated that increasing h, the total number of feature subsets, improved model accuracy, and the marginal gains in accuracy decreased as h became larger. This pattern suggested that although larger values of h led to better classification results, the benefits of adding more subsets grow smaller, eventually leading to a stabilization of performance at higher values of h.

When the number of non-informative features was relatively small, such as 100 or 200, the performance of RkCNN showed relative insensitivity to changes in h, maintaining stable accuracy across a broad range of h values. Conversely, in circumstances with 500 and especially 1,000 non-informative features, classification accuracy improved significantly as h increased. This was due to the larger pool of candidate subsets, which provided a higher probability of selecting subsets enriched with informative features. This trend highlighted that beyond a certain point, further increases in h did not substantially boost performance, as the accuracy plateaued.

The variation in r (200, 300, 400) exhibited similar patterns across all scenarios, suggesting that while a very large r is not necessary for achieving optimal performance, maintaining a sufficiently large r ensures efficient use of computational resources without redundant calculations. Overall, the analysis underscores the importance of carefully adjusting h and r according to the noise level in the data to optimize the accuracy of the RkCNN model.

Comparison

We compare RkCNN with several nearest-neighbor-based methods, including kNN, kCNN, and random k nearest neighbor (RkNN) and the ensemble of subset of k nearest neighbor (ESkNN). The misclassification rates under different numbers of non-informative features are presented in Table 1 and the corresponding average runtimes of ensemble methods are presented in Fig. 6. The tuning parameters were fixed at k = 3 and r = 200 through the comparison. For ESkNN, all other tuning parameters were set according to the recommendations in Gul et al. (2018). Given the diversity in feature numbers across datasets, we set the value of m for three ensemble methods associated with the feature size, specifically m=q.

Table 1 Misclassification rate for the methods on datasets with 20 informative features and different numbers of non-informative features.

Num of feature	kNN	kCNN	RkNN	ESkNN	RkCNN	
20	0.070	0.047	0.103	0.208	0.083	
20+50	0.088	0.070	0.055	0.190	0.042	
20+100	0.180	0.190	0.123	0.165	0.057	
20+200	0.135	0.125	0.178	0.188	0.047	
20+500	0.285	0.280	0.350	0.293	0.073	
20+1000	0.283	0.298	0.420	0.333	0.113	

Figure 6 Average runtime for different ensemble methods.

From the results in Table 1, kNN and kCNN performed competitively when only informative features were presented as shown in the first row of the table. Under these conditions, kCNN exhibited the lowest error rate among the individual methods, and RkCNN showcased the lowest error rate among the ensemble methods. This was expected since the effectiveness of feature utilization is maximized in the absence of noise. When non-informative features were included, RkCNN outperformed all other methods in terms of classification error rate. The error rate increased as more non-informative features were introduced into the dataset, yet RkCNN remained robust against this increase in noise. Among the three ensemble methods compared, RkCNN consistently outperformed both kCNN and ESkNN in all scenarios.

Figure 6 presents the runtimes of RkNN, ESkNN and RkCNN at r = 200. For all ensemble methods, runtimes increased with the number of features. Among the methods, RkNN exhibited the fastest runtimes compared to ESkNN and RkCNN. The runtime of RkCNN was affected by the choice of the parameter h. When h = r, RkCNN’s runtime was comparable to that of RkNN. However, when h = 3r, RkCNN’s runtime increased due to the additional computations for separation scores.

The performance variations across different numbers of non-informative features illustrated RkCNN’s robustness, leading us to further investigate the impact of the spread parameter ω on classification accuracy. The misclassification rates for different values of ω are presented in Table 2, offering insights into how each method copes with varying levels of entropy or uncertainty in class distributions.

Table 2 Misclassification rate for the methods on datasets with 20 informative features, varying numbers of non-informative features, and different values of ω.

	200 non-informative features	
ωk	kNN	kCNN	RkNN	ESkNN	RkCNN	
	1	3	1	3	1	3	1	3	1	3	
1	0.188	0.135	0.188	0.125	0.175	0.178	0.188	0.188	0.068	0.055	
3	0.267	0.272	0.267	0.255	0.188	0.233	0.235	0.215	0.113	0.157	
5	0.295	0.303	0.295	0.310	0.115	0.188	0.205	0.198	0.095	0.162	
10	0.305	0.340	0.305	0.353	0.080	0.100	0.090	0.178	0.035	0.125	
15	0.320	0.348	0.320	0.370	0.042	0.065	0.073	0.140	0.020	0.093	
	500 non-informative features	
1	0.323	0.285	0.323	0.280	0.338	0.338	0.262	0.248	0.085	0.057	
3	0.323	0.380	0.323	0.350	0.368	0.365	0.305	0.298	0.127	0.150	
5	0.335	0.370	0.335	0.370	0.280	0.308	0.225	0.223	0.140	0.140	
10	0.373	0.385	0.373	0.397	0.193	0.210	0.140	0.111	0.057	0.110	
15	0.360	0.400	0.360	0.413	0.123	0.155	0.090	0.113	0.042	0.080	
	1,000 non-informative features	
1	0.405	0.283	0.405	0.298	0.377	0.428	0.305	0.248	0.162	0.127	
3	0.382	0.360	0.382	0.308	0.402	0.430	0.360	0.300	0.195	0.215	
5	0.387	0.382	0.387	0.370	0.363	0.392	0.275	0.250	0.165	0.190	
10	0.363	0.395	0.363	0.443	0.285	0.333	0.170	0.167	0.125	0.135	
15	0.373	0.392	0.373	0.433	0.235	0.280	0.095	0.118	0.070	0.105	

As ω increased, particularly to 3 or 5, the ensemble methods exhibited their highest misclassification rates, indicating significant uncertainty in class distributions. At high levels of class overlap, nearest-neighbor-based approaches tended to perform well at k = 1, which is consistent with the results of García, Mollineda & Sánchez (2008). However, an interesting trend emerged where further increases in ω led to better classification results, suggesting a threshold beyond which the clarity of class separability enhances the models’ performance. RkCNN consistently outperformed other methods in all simulated cases, demonstrating superior robustness to both increased numbers of non-informative features and varied ω values.

This pattern was particularly notable in environments with higher noise levels. When 500 and 1,000 non-informative features were present, RkCNN’s advantage became more pronounced, confirming its effectiveness in managing complex scenarios with significant class overlap and high noise levels. Although the error rates generally increased with more non-informative features, RkCNN’s performance remained comparatively stable. This stability underscored RkCNN’s potential as a reliable tool for applications where data conditions were challenging.

Throughout this section, the comparative analyses and simulation studies have consistently highlighted RkCNN’s performance under various testing conditions. From managing datasets with large amounts of non-informative features to coping with increased entropy in class distributions, RkCNN proved to be effective in high-dimensional data that contain noisy features.

Experimental study on gene expression data

To evaluate and compare the performance of the proposed model, we conducted experiments using real-world gene expression datasets specifically designed for cancer microarray data classification analysis, as referenced in Mramor et al. (2007). These datasets are accessible for download at https://file.biolab.si/biolab/supp/bi-cancer/main-2.html. Table 3 summarises some statistics of each dataset including the number of observations, the number of features, the number of classes, and the class distributions. Some datasets, such as Tumor2 and Tumor3, originate from the same study but define their class labels differently, resulting in variable class counts. The datasets are predominantly high-dimensional, with feature counts ranging from 2308 to 22283, and most include no more than 100 observations, with some exhibiting significant class imbalance.

Table 3 Data summary of gene expression datasets used in the experiments.

	Instances	Features	Classes	Class distributions	
MLL	72	12,533	3	24/20/28	
SRBCT	83	2,308	4	29/11/18/25	
DLBCL	77	7,070	2	58/19	
Prostate	102	12,533	2	50/52	
LUNG	203	12,600	5	139/17/6/21/20	
LEU	72	5,147	2	42/25	
AML	54	12,625	2	28/26	
Childhood2	110	8,280	2	50/60	
Childhood4	60	8,280	4	13/21/16/10	
Breast Cancer	24	12,625	2	14/10	
Breast Colon	52	22,283	2	31/21	
Bladder	40	5,724	3	10/19/11	
CML	28	12,625	2	12/16	
Tumor2	23	9,945	2	11/12	
Tumor3	23	9,945	3	11/3/9	
gastric2	30	4,522	2	8/22	
gastric3	30	4,522	3	8/5/17	
LL2	19	15,434	2	9/10	
LL3	29	15,434	3	9/10/10	
lungCancer	34	10,541	3	17/8/9	
medulloblastoma	23	1,465	2	10/13	
prostate cancer	20	12,627	2	10/10	
braintumor	40	7,129	5	10/10/10/4/6	
glioblastoma	50	12,625	4	14/7/14/15	

A comparative analysis is made between the proposed RkCNN method and various well-known classifiers, including kNN, kCNN, and RkNN. We also include the random forest (RF) (Breiman, 2001) and support vector machine (SVM) (Cortes & Vapnik, 1995) in the model comparison. For all nearest-neighbor-based methods, we tuned the model parameters using a grid search with k = 1, 2 and m = 20, 50, 80 for RkNN and RkCNN. We fix r = 300 for RkNN and RkCNN, and h = 900 for RkCNN. RF and SVM were implemented using Python’s scikit-learn package. The classification accuracies were evaluated using leave-one-out cross-validation (LOOCV). Considering imbalanced class distributions, we also used the Matthews correlation coefficient (Matthews, 1975 MCC). The metric incorporates the full confusion matrix, providing a robust metric for reflecting classifier performance across both majority and minority classes. MCC values range from −1 to 1, where a higher value indicates better performance. An MCC of 1 denotes perfect classification, 0 indicates random prediction, and −1 reflects complete disagreement between predicted and actual values.

Tables 4 and 5 demonstrate that RkCNN outperformed all nearest-neighbor-based methods in 20 and 18 out of 24 datasets in terms of accuracy and MCC, respectively. In addition, RkCNN showed comparable performance to RF and, on average, achieved the highest LOOCV accuracy and MCC, as well as the best rankings on both metrics. These results validate that RkCNN not only competes effectively with traditional machine learning algorithms but also excels in addressing the complexities and variabilities inherent in gene expression data.

Table 4 LOOCV accuracy for each method across the gene expression datasets.

‘Average accuracy’ represents the average accuracy of each method. ‘Average rank’ represents the average of the method’s ranking across datasets.

Dataset	kNN	kCNN	RkNN	RkCNN	RF	SVM	
MLL	0.819	0.819	0.861	0.889	0.931	0.903	
SRBCT	0.855	0.855	0.904	0.952	1	0.964	
DLBCL	0.831	0.870	0.896	0.909	0.883	0.818	
Prostate	0.833	0.853	0.882	0.892	0.902	0.873	
LUNG	0.887	0.911	0.892	0.901	0.926	0.906	
LEU	0.699	0.849	0.781	0.863	0.973	0.877	
AML	0.593	0.556	0.611	0.556	0.648	0.537	
Childhood2	0.673	0.755	0.755	0.782	0.818	0.745	
Childhood4	0.367	0.450	0.450	0.467	0.400	0.450	
Breast Cancer	0.833	0.750	0.792	0.833	0.833	0.875	
Breast Colon	0.923	0.923	0.904	0.923	0.942	0.808	
Bladder	0.725	0.725	0.850	0.925	0.875	0.825	
CML	0.429	0.500	0.500	0.500	0.464	0.464	
Tumor2	0.870	0.870	0.913	0.913	0.913	0.609	
Tumor3	0.609	0.783	0.652	0.739	0.783	0.565	
gastric2	0.933	1	1	1	0.933	0.833	
gastric3	0.767	0.733	0.833	0.833	0.767	0.700	
LL2	1	1	1	1	0.947	0.947	
LL3	0.966	0.966	0.931	0.966	0.897	0.897	
lungCancer	0.618	0.618	0.676	0.735	0.735	0.588	
medulloblastoma	0.565	0.565	0.565	0.609	0.304	0.478	
Prostate cancer	0.600	0.700	0.700	0.700	0.700	0.600	
braintumor	0.725	0.725	0.700	0.700	0.725	0.675	
glioblastoma	0.580	0.580	0.720	0.740	0.760	0.640	
Average accuracy	0.738	0.766	0.782	0.805	0.794	0.732	
Average rank	4.583	3.646	3.271	2.375	2.583	4.542	

Table 5 LOOCV MCC for each method across the gene expression datasets.

‘Average accuracy’ represents the average accuracy of each method. ‘Average rank’ represents the average of the method’s ranking across datasets.

Dataset	kNN	kCNN	RkNN	RkCNN	RF	SVM	
MLL	0.744	0.744	0.771	0.853	0.896	0.854	
SRBCT	0.801	0.801	0.868	0.918	0.984	0.950	
DLBCL	0.502	0.664	0.787	0.707	0.585	0.460	
Prostate	0.667	0.706	0.745	0.786	0.786	0.747	
LUNG	0.763	0.818	0.785	0.786	0.839	0.808	
LEU	0.287	0.664	0.486	0.615	0.910	0.734	
AML	0.182	0.113	0.267	0.151	0.262	0.068	
Childhood2	0.358	0.507	0.485	0.560	0.578	0.491	
Childhood4	0.100	0.223	0.223	0.256	0.203	0.226	
Breast	0.657	0.543	0.580	0.676	0.657	0.742	
Breast	0.841	0.841	0.841	0.841	0.880	0.629	
Bladder	0.595	0.595	0.689	0.806	0.884	0.741	
CML	−0.108	−0.043	0.132	−0.021	−0.344	−0.300	
Tumor2	0.742	0.768	0.840	0.840	0.763	0.233	
Tumor3	0.350	0.641	0.414	0.641	0.391	0.228	
gastric2	0.853	1.000	1.000	1.000	0.829	0.553	
gastric3	0.616	0.616	0.707	0.645	0.666	0.473	
LL2	1.000	1.000	1.000	1.000	0.900	0.900	
LL3	0.950	0.950	0.950	0.950	0.950	0.856	
lungCancer	0.369	0.369	0.475	0.512	0.742	0.344	
medulloblastoma	0.115	0.115	0.115	0.115	−0.250	−0.271	
prostate	0.218	0.436	0.314	0.436	0.503	0.204	
braintumor	0.655	0.655	0.655	0.655	0.626	0.595	
glioblastoma	0.437	0.437	0.619	0.620	0.591	0.511	
Average MCC	0.529	0.590	0.615	0.639	0.618	0.491	
Average rank	4.563	3.604	3.063	2.438	2.813	4.521	

Overall, the proposed RkCNN method demonstrates leading or high rankings across the microarray datasets. When other methods were in the lead, RkCNN tended to be the second-best method. We also conducted pairwise Wilcoxon test (Wilcoxon, 1945) to assess whether the differences in rankings between the methods were statistically significant. Each test was performed as a one-sided test with a 5% significance level. According to the Wilcoxon test results, RkCNN ranked significantly better than the other methods in both accuracy and MCC, except RF, with no significant difference in ranking between RkCNN and RF(p-value = 0.549 in accuracy and 0.292 in MCC).

Findings in this section underscore the efficacy of RkCNN in practical applications, demonstrating its classification performance in scenarios characterized by high-dimensional data. The adaptability and superior performance of RkCNN suggest its suitability for complex classification tasks where accurate and reliable predictions are crucial.

Conclusion

In this article, we presented RkCNN, an innovative ensemble classification method that aggregates kCNN classifiers generated from random feature subsets. We also proposed the separation score metric to weigh class probability estimates derived from individual kCNN models. Comprehensive simulations were conducted to illustrate the efficacy of separation scores in enhancing model performance. Additionally, we undertook an in-depth exploration of the impact of each parameter on RkCNN’s performance, further deepening our comprehension of its functional mechanisms.

The simulation and experimental studies demonstrated that RkCNN outperforms other nearest-neighbor-based ensemble approaches, showcasing its effective handling of complex, high-dimensional data environments. Although the parameter tuning process is experimental within this article, other approaches such as adaptive tuning or theoretical analysis deserve further investigation.

While RkCNN filters random subsets of features based on their separation scores, it does not perform feature selection. However, the concept of ranking random subsets could potentially be adapted to filter individual features, thereby introducing a feature selection approach. This potential extension will be addressed in future work.

Beyond its application to gene expression data, the versatility of the proposed method suggests it could be effectively deployed in the analysis of other high-dimensional datasets, which are often characterized by a large proportion of non-informative features. This applicability potential makes RkCNN a promising tool in a wide range of data-intensive fields.

Supplemental Information

Supplemental Information 1 Implementing the proposed RkCNN algorithm in Python

Supplemental Information 2 Code

Additional Information and Declarations

Competing Interests

Author Contributions

Data Availability

The authors declare there are no competing interests.

Jiaxuan Lu performed the experiments, analyzed the data, performed the computation work, prepared figures and/or tables, authored or reviewed drafts of the article, and approved the final draft.

Hyukjun Gweon conceived and designed the experiments, authored or reviewed drafts of the article, and approved the final draft.

The following information was supplied regarding data availability:

The datasets used are available at: https://file.biolab.si/biolab/supp/bi-cancer/main-2.html.

The python code files are available in the Supplemental Files and Zenodo: aljers. (2024). aljers/RkCNN: RkCNN Release Version 1.0 (Main). Zenodo. https://doi.org/10.5281/zenodo.13855998.

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
