# Peer review of "Random k conditional nearest neighbor for high-dimensional data"

_PeerJ Computer Science, doi:10.7717/peerj-cs.2497_

## Round 0.1 · original submission · Major Revisions

Dear authors,

Please take into account all the comments of the reviewers, some of them require new experiments that must be launched.

Regards

Reviewer 1 ·

Basic reporting

Minor Points for Improvement:

1 - For better understanding, explain what 'm' represents in the figure captions.

2 - All figures are included at the end of the document. If the journal's formatting allows it, it would be better to include them within the text, as is done with the tables, to facilitate reading.

3 - A table indicating the characteristics of the artificial datasets used is missing, similar to Table 3, which describes the characteristics of the real data.

4 - Explain what the "response" column represents in Table 3.

5 - Sections are referred to with numbers (e.g., "Section 2.2"), but the sections are not numbered. This should be corrected if the journal’s formatting allows it.

Experimental design

Major Points for Improvement:

1 - Evaluation Metric: The paper uses accuracy as the evaluation metric, which is sensitive to imbalanced data. It would be more appropriate to use more robust metrics such as the Matthews Correlation Coefficient, F1 score, Area Under the Curve, etc.

2 - Statistical tests: When comparing results, it is always advisable to use a statistical test, such as the Wilcoxon test, to verify that the differences are statistically significant.

3 - If possible, it would be helpful to include information on the algorithms' execution time.

Validity of the findings

no comment

Additional comments

The paper's strengths include a long list of wide datasets and the utilization of LLOCV, which provide robust validation of the proposed method. Furthermore, the authors provide the code for the algorithm, facilitating result replication and method verification by other researchers.

Cite this review as

Reviewer 2 ·

Basic reporting

This paper introduces a novel strategy called Random k Conditional Nearest Neighbor (RkCNN) for classifying high-dimensional data, aiming to reduce the classification error caused by noise features. The proposed approach involves four key steps: (1) randomly selecting subsets from the full feature set, (2) evaluating and filtering these subsets based on their classification power, (3) constructing classifiers on the selected subsets, and (4) aggregating predictions from these classifiers to produce the final prediction. The paper also introduces a new score metric to assess the classification power of feature subsets, making it easier to filter out noise features. Extensive experiments, both simulated and on real-world datasets, are provided to validate the method.

Strong Points:
S1. Novel Metric for Feature Selection: The introduction of a novel score metric, which is the ratio of between-group variance to within-group variance, is a significant contribution. This metric effectively identifies informative features and avoids noise when constructing classifiers. It is both reasonable and straightforward to implement.
S2. Use of Multiple Classifiers: The paper innovatively uses multiple classifiers by selecting informative feature subsets from multiple randomly sampled candidates. This approach increases the likelihood of capturing informative features, which is a strong point of the methodology.
S3. Weighted Aggregation of Predictions: The strategy of aggregating predictions from all classifiers using weights based on their classification power is simple yet effective. This ensures that each classifier contributes appropriately to the final prediction, improving classification accuracy.

Weak Points:
W1. Lack of Comparison with Existing Methods: The paper does not provide sufficient comparison between the proposed feature selection approach and existing dimensionality reduction methods, such as PCA. A discussion on the advantages of the proposed method over these classical approaches would strengthen the paper.
W2. Alternative Classifier Construction Strategy: The paper could discuss the possibility of constructing an effective classifier using a more straightforward strategy. For example, by computing the classification power of each single feature, selecting informative features to form one feature subset, and then building a single classifier. This approach could reduce the uncertainty introduced by random feature selection and lower computational costs.
W3. Unexplained Experimental Results: Some experimental results appear unreasonable, particularly in Table 2 where, for 200 non-informative features, an increase in k from 1 to 3 results in a higher misclassification rate for RkCNN. Typically, a higher k should decrease the misclassification rate by providing more information. The paper should include a discussion to explain these results.

Experimental design

Please refer to the weak points in the basic reporting

Validity of the findings

no comment

Additional comments

This paper presents a novel and promising approach to classifying high-dimensional data. While the methodology is sound, addressing the weak points—particularly the comparison with existing methods and the discussion of alternative strategies—would significantly enhance the paper’s contribution to the field. I recommend the paper for acceptance with minor revisions.

Cite this review as

---

## Round 0.2 · accepted · Accept

I confirm that the paper is now Acceptable.

Reviewer 1 ·

Basic reporting

no comment

Experimental design

no comment

Validity of the findings

no comment

Additional comments

The proposed changes have been successfully resolved.

Cite this review as

Reviewer 2 ·

Basic reporting

Thank you for submitting the revised version of your paper. I appreciate the thorough revisions, and I am pleased to see that all of my concerns have been effectively addressed.

Experimental design

Thank you for submitting the revised version of your paper. I appreciate the thorough revisions, and I am pleased to see that all of my concerns have been effectively addressed.

Validity of the findings

Thank you for submitting the revised version of your paper. I appreciate the thorough revisions, and I am pleased to see that all of my concerns have been effectively addressed.

Cite this review as